# Accelerating Open Science for AI in Heliophysics

**Dolores Garcia**
Trillium
London, UK
dolo@trillium.tech

**Paul J. Wright**
Wright AI Ltd
Leeds, UK LS2 7HZ
paul@wrightai.com

**Robert Jarolim**
University of Graz
Graz, Austria
robert.jarolim@uni-graz.at

**Mark C. M. Cheung**
CSIRO
Marsfield, NSW 2122, Australia
mark.cheung@csiro.au

**Meng Jin**
Lockheed Martin Solar
Astrophysics Lab
Palo Alto, CA 94304
jinmeng@lmsal.com

**James Parr**
Trillium
London, UK
james@trillium.tech

## Abstract

Rarely are Artificial Intelligence (AI) projects packaged in a way where scientists and non-AI specialists can easily pick up advanced Machine Learning (ML) workflows. Similarly, AI engineers are not always able to contribute meaningfully to a science domain without being provided with useful application context or analysis-ready data. Because of this–and other factors–applied AI research often stalls at the research paper stage, where the often complex logistics of replicating and building on the work of others impedes substantive progress. A state of affairs has been identified by the community as 'Reproducibility.' (1,500 scientists lift the lid on reproducibility [4]). Potential gains in AI are therefore hampered by the "expertise gap" between ML specialists and domain scientists. Moreover, the reputation of AI as a transformative tool for science is somewhat belated due to the lack of deployed, trusted solutions in the wild–as projects struggle to migrate from mid-TRL (Technology Readiness Level) to high TRL. Another key concept is that AI projects are never really finished. Improvements can be made in both the model choice (the selection of which improves annually) and training data–the latter often being the key actor in improving outcomes. In this paper we present the learnings for a study conducted for the NASA Heliophysics Division and UCAR to tackle findings informed by the 2021 NASA Science Mission Directorate AI Workshop [16], showcasing best practice in the adoption of trusted and maintained open science in AI for Heliophysics and scaling lower TRL applications to higher TRLs. We also present an example of rapid derivative Heliophysics research conducted by a non-subject matter expert, showing the value of these kinds of open science approaches.

## 1 Introduction

The recorded data from our closest star keeps growing, collected through missions such as the Solar Dynamics Observatory (SDO; [18]). At the same time, new Artificial Intelligence (AI) technologies are becoming mainstream, advancing challenging problems such as object detection, automated decision making, and deducing relations from high-dimensional data. AI is already unlocking the potential of these large datasets in Heliophysics, and accelerating scientific discoveries [6]. However, with the increasing availability of tools and data, there is an increase in the misuse of these tools and failures that can lead to invalid scientific results. For example in the SDO dataset, the images (obtained by different instruments) need to be spatially and temporally adjusted, and in the case of

2022 Trustworthy and Socially Responsible Machine Learning (TSRML 2022) co-located with NeurIPS 2022.

the Atmospheric Imaging Assembly (SDO/AIA), these also need to be corrected for time-dependent degradation. Using level-1 data, i.e. minimally-processed instrument data (see § 5 of [15] for a discussion of Level 0, 1, and 1.5 data), openly available without accounting for this nuance can lead to incorrect results. This need for domain understanding of data limits the broad application of AI to scientific research, while the prevalence of avoidable errors reduces the trust in the results and the use of AI. In light of these scenarios, the community needs to establish best practices for data curation and annotation to increase the trustworthiness and openness of AI in science applications [7]. Reproducibility and best practices for simplifying derivative AI experiments have been a consistent challenge for the community. Platforms for increasing the reproducibility and creating an engaged community are still nascent. Some tools are currently available to share code and datasets and improve reproducibility, such as Github, PaperswithCode [1], and Pangeo [2]. However, these platforms are not specific for scientific applications and do not actively loop in domain experts in data curation. Often they do not hold requirements for the content that is uploaded, for example regarding detailed documentation or code annotation. They do not provide access to compute for rapid experimentation or actively promote a community around the component tools that may simplify the production of derivative research.

The platform used for this study, SpaceML (spaceml.org), aims to create an open science resource to host space science datasets and data products, pipelines and tools for ML with Heliophysics applications, and additionally cultivate an active community of data product curators. It differs from other available tools in that it facilitates the maintenance of scientific AI-ready data, models and tools, establishes requirements for shared tools (documentation, notebooks, versioning and verification by domain experts) and in that it actively engages the Heliophysics community to properly verify, improve and transfer the knowledge that is created. In the following sections, we discuss techniques to increase reproducibility and create a community-driven continuous optimization that leads to larger scientific impact, and show the results of such techniques in two case studies.

## 2 SpaceML

In this section, we discuss the techniques developed for preparing and sharing datasets, pipelines, and tools for Heliophysics on the SpaceML platform. We also discuss how these techniques improve reproducibility and access to the resources in the Heliophysics community.

**Datasets**    The SpaceML platform hosts expert-informed AI-ready datasets.

Preprocessing – The processing stages for these datasets are: (1) obtain raw data from available sources e.g NASA public repositories such as Stanford's Joint Science Operations Center (JSOC), (2) pre-process the scientific data with subject matter expertise to ensure consistency and remove the need for instrument-specific knowledge, and (3), format the data to be ML ready. The second step is of key value for accelerating AI in scientific applications as it requires domain expertise which can be sometimes neglected in ML applications. For example, this step can include instrument calibration to compensate for degradations that are time-dependent. Skipping this step can lead to costly mistakes and can introduce biases. This workflow results in AI-ready datasets that have been verified by experts and are ready for use by the community to develop AI Heliophysics pipelines.

Dataset – Each dataset includes a dataset card, with the format described in [14], that documents the project information (total size, format, owner, versioning, context, intended scientific use) and implicit knowledge such as modeling assumptions, biases, etc. Each dataset also includes the code of the preprocessing steps and a Google Colab notebook with a guide for download, use of the dataset and examples of querying. This step is key to ensure that the datasets are of value to the community.

Access – The datasets are openly accessible and are hosted on Google Cloud. For large datasets, a subset of the larger dataset is prepared for users to download and test. This removes the large memory requirement to host the full dataset locally.

**MLOps**    The SpaceML platform provides a set of data management tools that are reusable across projects. For example tools such as data-downloaders for popular public datasets, such as NASA GIBS Worldview [3], or data loaders for some of the hosted datasets that allow the datasets to be batched and read by an ML model. These tools are pieces of larger pipelines that are general and can be reused for multiple applications. By extracting these pieces and making the functions available

we accelerate the process of developing ML for scientific purposes, abstracting the implementation and unlocking time for scientific value. These tools also enable non-ML specialists to develop applications and are examples of good practices for the community. By creating these tools we move away from the tendency of rushed processes, duplicated effort, and misuse of the datasets.

**Pipelines** The SpaceML platform currently provides six ML pipelines for Heliophysics applications and another five for other scientific purposes such as Earth Science. For example, it includes a pipeline to create super-resolved maps of the solar magnetic field [11], and a pipeline to track the geo-effectiveness of solar storms [20]. These pipelines are curated and well documented, and bring the following benefits to the Heliophysics community: (1) they provide benchmarks for different Heliophysics problems (such as super-resolution) that are versioned, easily accessible by the community, and verified by domain experts, (2) they provide examples of how to use ML for Heliophysics together with best practices, which can be of high value for non-expert ML users and to introduce Heliophysics applications, and (3) they provide trustworthy results that showcase the feasibility of AI for scientific purposes, for example by showcasing uncertainty estimations calculations for different purposes. As in the datasets section, the ML pipelines also include a "model card" that documents the information and the implicit knowledge of the pipeline. Additionally, each pipeline includes a repository where all the code can be downloaded and a Google Colab notebook. This notebook demonstrates how to download and read the necessary data, the model setup, the hyperparameters, and a discussion on the results of the model. These notebooks increase the ease of reproducibility, as it is an environment where the dependencies are already installed, and the results from the study can be easily obtained without having to download the repository, the data and setting up the dependencies. This mechanism allows other researchers to directly access the results of the pipeline and continue iterating without tedious delays due to implementation.

The purpose of SpaceML hosting the projects is twofold. First, we would like to centralize the storage of all project information, and second, we want to encourage derivative research by allowing the use of the data and techniques currently housed within the project. SpaceML can do this by creating an open-source community to encourage knowledge sharing. We expect that when further research results in the development of a new tool, the enhancement of the model or the creation of a variation of the dataset, these will be registered with SpaceML. Additionally, all new derivative datasets will be uploaded using versioning guidelines to correlate tools, datasets, and research applications.

**Communities** SpaceML aims to foster an inclusive and collaborative community that can develop cutting-edge research to achieve a large scientific impact. This community can benefit from the curated tools and deployment to build on their research but also contribute to the platform. The activities being developed are the following. Visitors are invited to join the official Slack channel for SpaceML. Registered members will also receive quarterly newsletters updating them on developments in current SpaceML projects, news, new projects and SpaceML events. SpaceML has recently launched two initiatives for knowledge transfer. 1) Blogs: SpaceML blogs focus on multiple facets including highlighting user journeys, mentor experiences, project case studies and introduction to new tools and techniques. These blogs are being published under the SpaceML medium account, and links to each blog are also being housed on the SpaceML webpage. 2) Speaker series: monthly seminars, composed of a technical presentation about one of the tools-pipelines-datasets followed by a discussion section. These are recorded and shared online. After the release of an episode, SpaceML follows up with SpaceML surgeries/office hours sessions. These sessions serve as a means to directly interact with the subject matter expert. These surgeries will be hosted live as a QA on Slack to allow community engagement and enable researchers to come together.

## 3   Simplifying Derivative Research: Case Study

In this section we showcase two case studies in which SpaceML has accelerated research in Heliophysics. The first case study showcases the addition of a new tool to the SpaceML platform, an AI-ready dataset for the SDO. The second case study shows how the SpaceML platform helps a PhD level researcher accelerate their research.

### 3.1 SDOML Dataset

The SDO is designed to help us understand the Sun's influence on Earth and Near-Earth space by studying the solar atmosphere on small scales of space and time in many wavelengths simultaneously. The mission launched in 2010 has been monitoring the Sun's activity and delivering scientific data from three instruments: The Atmospheric Imaging Assembly (AIA; [15]), the Helioseismic and Magnetic Imager (HMI; [19]) and the Extreme UltraViolet (EUV) Variability Experiment (EVE; [22]). The tens-of-petabytes of freely available Level-1 scientific data have been widely used by the community (refereed in over 5000 publications). However, with an increasing number of independent groups applying ML to SDO data, this naturally results in varying levels of data preparation and preprocessing which can often be inappropriate for this scientific data, and complicate model comparisons. While the SDO spacecraft is reliable and the observations consistent, there are known artifacts. For example, the apparent size of the Sun varies around 3% due to the eccentric orbits of the Earth around the Sun, the exposure time varies, and other anomalies occur when the cameras are offline or due to spacecraft off points from the Sun's center for calibration sequences. The goal of this case study was to release a standardized ML-ready dataset that was appropriately prepared for a diverse set of problems. One of the main hurdles that researchers face with SDO data is the sheer volume of data. To reduce the size of this data, the SDOML dataset uses down-sampled images (from $4096 \times 4096$ to $512 \times 512$ pixels), before being reduced temporally from 12 second to 6 minute cadence (the Sun rotates once every $\sim$27 days).

The initial version of the SDOML dataset was released in 2019 (v1.0; [10]). Here, images were grouped by wavelength and time and saved as Numpy [12] `.npz` files, amounting to a total of 7TB and stored on the Stanford Digital Repository. Currently, the dataset is at v2.0 (see [23] for a video overview). The data is now hosted on the Google Cloud Platform in the `.zarr` format [17], and has been updated to include the full FITS[1] header information as metadata. These changes increase not only the amount of data available, but the ease of data access, enabling more scientific applications and data exploration. Furthermore, the degradation correction factor is now saved within the metadata so that it is possible to revert to raw data if needed. Importantly, this new data format permits on-cloud computing and model training without downloading the dataset locally, which will greatly facilitate the ML studies using the dataset. This version further incorporates a suite of protocols, metrics, and baseline models, lowering the barrier of using Heliophysics data for non-Heliophysics ML researchers who may be unfamiliar with domain-specific nuances.

#### 3.1.1 SDO/AIA Auto-calibration

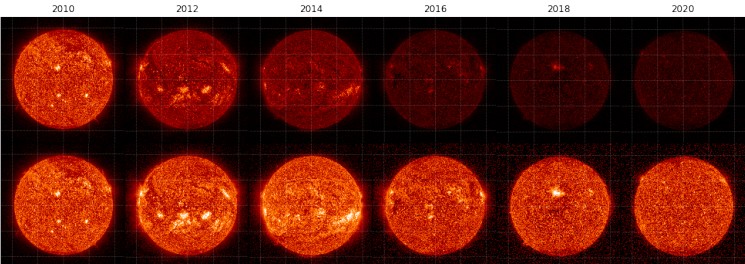

Figure 1: Top (left to right) shows the bi-yearly observation of SDO/AIA 304 Å data as observed from 2010 through to 2020, plotted on the same colorscale. The observed dimming is a result of the degradation over time. Bottom: as top, after correction by the SDO/AIA auto-calibration model [9].

The SDO/AIA auto-calibration model [9] builds upon the SDOML dataset to provide a real-time correction for the known time-dependent degradation that is seen in EUV filters[2]. In this work, the authors used a convolutional neural network (CNN) that takes as input either a single SDO/AIA channel, or multiple channels, outputting the degradation factor (or factors for a multi-channel

---

[1]FITS is a file format used for the transport, analysis, and archival storage of scientific data sets (data and header information) [21]. See `https://fits.gsfc.nasa.gov/`.

[2]Until 2014, SDO/EVE was able to provide spectral information relevant for calibration. Since 2014, calibration of SDO/AIA has been performed with sounding rocket flights, however, these flights are sparse (roughly once every two years), and degradation correction factors are interpolated between successive flights.

version). In the accompanying notebook, hosted on SpaceML, the user is guided through this project by reading and loading the SDO/AIA data (with no degradation correction present) before performing inference with the multi-channel model, and applying the degradation factors back to the original AIA data. Figure 1 shows bi-yearly observations of SDO/AIA 304 Å data as observed by SDO/AIA from 2010 through to 2020, plotted on the same colorscale pre- (top), and post-correction (bottom).

While this correction was outside the scope of the SDOML dataset, the SDO/AIA auto-calibration project (also hosted on SpaceML) has built upon the dataset to provide per-image correction for the degradation observed in EUV filters, crucial for studies such as the one discussed in the § 3.2.

### 3.2 Instrument-to-instrument (ITI) Pilot

In this pilot we want to showcase how SpaceML can accelerate Heliophysics research for scientists that become users of this tool. For the pilot, the scientist is a PhD-level researcher that develops ML applications for Heliophysics.

The researcher was working on the following problem: In solar physics, the long-term evolution of the magnetic field typically exceeds the lifetime of a single instrument and this limits the application of data-driven approaches to historical data samples. They developed a ML approach for domain translation between different instruments (Instrument-To-Instrument translation; ITI) [13] with the aim of providing a uniform data series of EUV observations from SDO/AIA, STEREO/EUVI [24] and SOHO/EIT [8]. However, their method of unpaired image translation, when applied to standard reduced SDO/AIA data, showed a sensitivity for insufficiently corrected device degradation, leading to differences between the calibrated series.

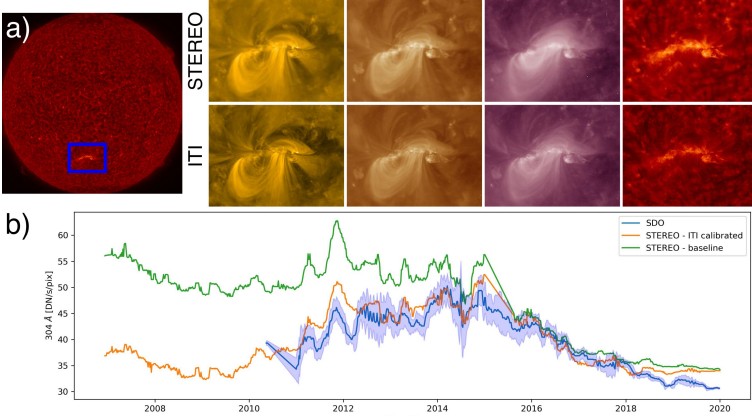

Figure 2: Intercalibration of STEREO and SDO observations using ITI and the SDO auto-calibration. a) Comparison of the original STEREO observation (top) and the ITI enhanced version (bottom) for the individual filtergrams (171, 195, 284, 304 Å). The calibration results from a feature dependent image translation based on the SDO image distribution. b) Average intensities of the reference SDO observations, the ITI enhanced observations with standard calibration (baseline), and ITI enhanced observations with improved calibration prior to the translation. The data is smoothed by monthly averaging, where the blue shaded area indicates one standard deviation of the SDO series. The prior calibration increases the agreement between the inter-calibrated data. (reproduced from [13])

The researcher took advantage of SpaceML by using the provided auto-calibration from [9] (see 3.1.1) to obtain a consistent calibration for the SDO/AIA data series and used the ITI framework to translate observations from STEREO/EUVI to the same domain. They showed that with the adjustment available in the SpaceML platform they could achieve an accurate calibration between the three instruments and that the comparisons of aligned observations demonstrated high perceptual quality and strong similarity to reference observations.

This result was achieved in three weeks, since the dataset preprocessing and auto-calibrations were available in the platform and well documented. The publicly available codes accelerated the implementation of the code adjustments (i.e., preprocessing). The researcher also benefited from the SpaceML community and engaged with the authors of the SDOML dataset [10] to discuss the

details of the auto-calibration. On one hand, this is an example of how SpaceML's openly available tools can accelerate the science turnaround time for a PhD level researcher. But also it shows how the community of SpaceML is of key importance for this goal.

The result of this pilot study is a data series covering uniform observations dating back to 1996, including simultaneous observations from multiple vantage points. This dataset paves the way towards a new generation of solar cycle studies of the solar EUV corona, contributes additional samples for data-driven methods and enables the application of automated methods that were developed specifically for SDO/AIA data to the full EUV data series without further adjustments.

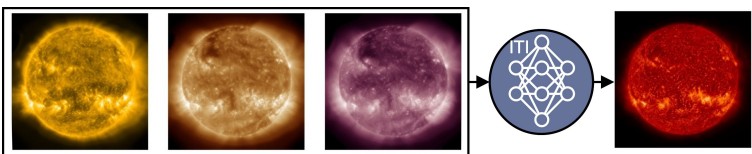

Figure 3: Example application of ITI for data synthesis. Image translation from three coronal EUV channels (171 Å, 193 Å, 211 Å), to the chromospheric 304 Å channel. The model is trained with the use of the SDOML dataset and online resources.

As presented in the § 3.1, SpaceML provides the SDO dataset [10]. Additionally, after obtaining a consistent calibration for the SDO/AIA series, the researcher used the SDOML dataset to provide example applications of ITI. Specifically, they showed: 1) a reconstruction of the chromospheric 304 Å channel based on coronal EUV observations as shown in Figure 3, i.e. similar to the STEREO to SDO image generation, now the network learns to generate a new channel. 2) An estimation of the line-of sight magnetogram (SDO/HMI) based on the EUV filtergrams. Given that the SDO dataset can be easily distributed, the models can be trained online using the Google Colab platform. These applications, together with the ITI pipeline and corrected calibration of the SDO/AIA band, are now available at SpaceML.

This pilot study showed that there are also possibilities for improvement. A unified data loader for the SDOML dataset could enable a faster integration with common deep learning frameworks. At the present stage this requires a custom implementation for the individual applications. Frequently researchers are not interested in the method itself, but only require the derived data products. Hosting the primary data products (e.g., calibration curve, enhanced images) allows researchers that have no experience with ML methods to also benefit from these methods. Countrarily, researchers that are primarily interested in the method would benefit from further interactive descriptions of model training, which could foster the application to similar problem settings.

## 4  Conclusions

**More than just sharing data:** SpaceML includes analysis-ready datasets, space science ML projects, and MLOps tools designed to fast-track existing AI workflows to new use-cases. The datasets and projects build on the six years of cutting-edge AI application completed by Frontier Development Lab (FDL) teams of early-career PhDs in AI/ML and multidisciplinary science domains in partnership with NASA, USGS, ESA, and FDL's commercial partners. SpaceML assures that hosted applications follow the same standard and are verified by domain experts. The direct access to the corresponding data sets and provided Google Colab notebooks, allow to easily utilize the applications. The SpaceML platform provides an important resource for discovering recent state-of-the-art deep learning applications in Heliophysics.

**Benchmarking Tools to TRL-7:** Sharing of analysis-ready data products accelerates the benchmarking of open applied AI pipelines and data products for rapid and easy assessment by the science and data science community. This can be thought of MLOps for AI efficacy and trust; with a narrower set of requirements than would be needed for a full generalizable MLOps suite with other researchers as the primary users. Another way of framing this is that despite earnest calls for progress in AI unless there are mechanisms for trust, deployed AI solutions and community-maintained Analysis Ready Data (ARD) will remain outliers, rather than the norm.

**Full Deployment to TRL-9+:** The ideal outcome of AI research is the full deployment of an operational system for a NASA, UCAR, or NOAA-style use-case. As discussed already, to be fully

trusted and compliant with pending plans for AI deployment, pipelines need to be developed with the principles explored here at their core: understandable, reliable, and explainable.

Making these projects available to the community in a way that simplifies onboarding, reproducibility, and derivative research for both constituencies is thus a key concept in promoting the use of trustworthy AI in the Federal Government.

**Acknowledgments**

This study was supported by UCAR and NASA Heliophysics Division. The authors would like to thank the SETI Institute, Lockheed Martin and the researchers from the Frontier Development Lab for the case studies used as building blocks for these ongoing investigations.

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

# A  Related work

## A.1  Platforms to share code

- GitHub (github.com) is one of the major platforms for developing, sharing, and versioning codes. GitHub repositories associated with scientific publications are primarily maintained by the researchers, and can strongly vary in their documentation, reproducibility and usage. Furthermore, the corresponding data for training and evaluation is typically not included due to storage limitations.

- Paperswithcode (paperswithcode.com) is a platform that provides ML papers, code, dataset, methods and evaluation tables. As in Github, these are associated with publications and are maintained by researchers. Therefore the same variations in documentation and reproducibility are present. Additionally, not all uploaded papers have their code or datasets available.

## A.2  Heliophysics for ML reproducibility educational resources

- The *Machine Learning, Statistics, and Data Mining for Heliophysics* e-book helioml.org [5] provides examples of how to use ML, statistics and data mining for Heliophysics datasets to help researchers increase their reproducibility.

## A.3  Raw data

- The Virtual Solar Observatory (VSO[3]) provides a large data repository from various sources of solar observations (e.g., SDO; the Global Oscillation Network Group, GONG; Parker Solar Probe, PSP), but domain knowledge is required for processing and working with specific datasets.

- Instrument archives are one of the primary sources for accessing data and provide information for processing the specific data (e.g., JSOC, Hinode[4], SWPC[5]). The individual archives are not centralized and have different interfaces, which makes it difficult to automatically discover and acquire large data sets.

# B  Terminology

**Technology Readiness Level**  The Technology Readinness Level concept, which defines the maturity of a technology, was extended for the specific case of Machine Learning in [14]. In [14] the authors define the framework and terminology spanning through prototyping, productization, and deployment of an ML system.

---

[3] https://virtualsolar.org/
[4] http://sdc.uio.no/sdc/
[5] https://www.swpc.noaa.gov/

