# OpenReview forum: "Accelerating Open Science for AI in Heliophysics"
_NeurIPS.cc/2022/Workshop/TSRML — TSRML2022_

### Official Review · Reviewer_8BoJ · 2022-10-18
**Review of Paper62 (recommend accept)**

**Overall Rating:** 7

**Summary:**

The authors propose a new platform "AI4Space" to improve the accessibility and usability of Heliophysics related data. The platform includes dataset download, MLOps, and ML pipelines. "AI4Space" addresses an important problem in the scientific community, which is dealing with a great influx of data and a need for AI pipelines. The authors demonstrate the usefulness of this platform through two case studies: one describing the addition of a new dataset and the other describing how a researcher would use the platform.

**Strengths:**

This work addresses an important problem at the intersection of AI/ML and science. While the work does not propose new research, it does provide a useful platform that will be able to support new research in the future and for that reason, I believe it would be relevant to present at this workshop. I thought that the platform as described in Section 2 seemed to be quite thorough in terms of addressing potential needs that a Heliophysics researcher may face when trying to use AI/ML systems.

**Weaknesses:**

Some of the terminology mentioned (but not defined) in this work may not be accessible to much of the audience at the workshop (e.g., level-1 data, TRL). I would appreciate the addition of some text that provides more context for such terms in the appendix.

While I found most of the paper easy to follow, I was confused by parts of the abstract. For example, the abstract mentions an "SMD AI Workshop", which is never referred to again in the main text.

**Overall Recommendation:**

I would recommend accepting this work and would suggest authors revise the submission based on the comments mentioned above.

**Review Confidence:**

2: The reviewer is willing to defend the evaluation, but it is quite likely that the reviewer did not understand central parts of the paper

---

### Official Review · Reviewer_7gU5 · 2022-10-19
**Good case study paper**

**Overall Rating:** 7

**Summary:**

This paper introduces AI4Space platform, a practical platform that accelerates the application of ML in heliophysics. Several practical challenges, blind spots, and corresponding solutions are presented.

**Strengths:**

- A practical and useful platform, AI4Space, is introduced in the paper. The platform incorporates several novel solutions, especially for normalizing data, that eases the wide application of modern DL techniques in heliophysics.

- Two case studies demonstrate the usefulness of the platform and the pipeline.

**Weaknesses:**

- Some part of the paper is a bit hard to follow by outsiders. For example, in Figure 3, how to transform across different channels? Also, the paper could conduct more empirical studies on the effectiveness of the proposed pipeline.

- The paper could highlight the blind spots and lessons learned during the development of the platform to bring takeaways for other AI4Science practitioners.

**Overall Recommendation:**

Overall, I believe it is a good paper that demonstrates how to increase the trustworthiness of AI in concrete applications through case study. The hands-on experience is valuable for the community.

**Review Confidence:**

3: The reviewer is fairly confident that the evaluation is correct

---

### Decision · Program_Chairs · 2022-10-23

**Decision:**

Accept

**Comment:**

Following the unanimous recommendations from reviewers, the submission is accepted.